# Effects of Long-Term Application of Earthworm Bio-Organic Fertilization Technology on Soil Quality and Organo-Mineral Complex in Tea Garden

**Huan Li** [1,2], **Yang Zhou** [1], **Huiling Mei** [3], **Jianlong Li** [4], **Xuan Chen** [1], **Qiwei Huang** [3], **Xinghui Li** [1,*] and **Jinchi Tang** [4,*]

1   International Institute of Tea Industry Innovation for "the Belt and Road", Nanjing Agricultural University, Nanjing 210095, China
2   Jiangsu Key Laboratory for Horticultural Crop Genetic Improvement, Institute of Leisure Agriculture, Jiangsu Academy of Agricultural Sciences, Nanjing 210014, China
3   College of Resources and Environmental Sciences, Nanjing Agricultural University, Nanjing 210095, China
4   Tea Research Institute, Guangdong Academy of Agricultural Sciences, Guangzhou 510640, China
*   Correspondence: lxh@njau.edu.cn (X.L.); tangjinchi@126.com (J.T.); Tel.:+86-25-8439-5182 (X.L.)

**Abstract:** Soil quality is crucial for plant productivity and environmental quality sustainability. Applying bio-organic fertilizer to achieve sustainable agriculture has become popular. Tea garden soil which had been fertilized for 12 years was chosen for the study, and soil quality and microaggregate composition were studied. The results showed that earthworm bio-organic fertilizer treatment could increase the indicators of soil's physical and chemical properties such as total carbon and total nitrogen in soil. Bio-organic fertilization technology could significantly increase the number and activity of soil microorganisms, and upgrade soil enzyme activity which was related to soil nutrients. Specifically, the activities of urease in soil were markedly enhanced due to the implication of bio-organic fertilizer. Additionally, SR-FTIR analysis revealed that clay minerals were connected as nuclei with the capacity to bind carbon, and that this interaction was aided by organic fertilization. Specifically, the replacement of chemical fertilizer with organic fertilizer can improve the ability of clay minerals and iron/aluminum/silicon oxides to protect aliphatic groups, polysaccharides and proteins. In conclusion, continuous organic amendments initialize a positive feedback loop for the maintenance of the organic–mineral complex in soils, which can contribute to enhanced soil organic carbon (SOC) storage. These results confirmed the feasibility of organic fertilizer for soil quality improvement in tea plantation ecosystems.

**Keywords:** soil quality; tea garden soil; bio-organic fertilizer; organo-mineral complex





## 1. Introduction

The tea plant (*Camellia sinensis* Kuntze) is a significant commercial crop in the world [1]. Fertilization, which affects soil structure and fertility and encourages crop development, is the most fundamental agricultural production management strategy used in the production of tea [2]. Many issues still exist in China's tea gardens today, such as improper fertilization practices and insufficient organic fertilizer application, which raise expenses, increase the danger of environmental contamination and also compromise the quality and productivity of the tea [3]. It is well known that the addition of organic fertilizer has beneficial effects on soil quality by improving the soil porosity and the soil organic carbon (SOC) content [4], and it contains all essential nutrients which are slowly released [5]. The addition of inorganic fertilizers to organic fertilizers (green manure, agricultural compost) stimulates soil biological activities by increasing plant biomass and each activity increases significantly [6]. Organic fertilizer has the characteristics of comprehensive nutrients and high availability, which can promote the breeding of soil microorganisms and improve

the ability of soil to maintain fertilizer and water [7], so it is conducive to accelerating the formation of soil aggregates and improving the physical and chemical properties of soil. Nutrient amendment may also affect soil aggregate stability and soil mineral availability via changing plant root distribution and soil pH [8]. A sustainable yield depends on soil structure, organic matter and nutrient cycling, which are functions of chemical, physical and biological properties. Soil properties based on biological and biochemical activities, such as soil enzymes, have been shown to respond to small changes in soil conditions, thereby providing sensitive information on small changes in soil quality [9]. The soil quality resulting from enzymatic activity is described and defined by enzymatic activity indicators and provides valuable information on soil fertility conditions. The biological activity of soil can be altered by changes in environmental conditions and pollutants, such as toxic substances such as heavy metals. These changes can destabilize soil systems, resulting in reduced soil accumulation [10].

In agroecosystems, earthworms are well-known "ecosystem engineers" seen as crucial to maintaining healthy soil and plant development [11]. Earthworms are the primary soil ecosystem engineers, facilitating the development of soil's physical structure (such as the generation of porosity and the macro-aggregation process) and associated soil [12]. Through their casting and feeding behaviors, earthworms in particular have been proven to change the nutrients that plants may use [13]. Makkar et al. revealed that vermicompost is being used as a component of organic farming, making it imperative to study the impact of vermicompost on the growth of different plant species [14]. Lubbers et al. showed in a long-term (750 days) mesocosm experiment that earthworms increase both $CO_2$ generation and the incorporation of carbon (C) into aggregate fractions, although the mineralization of C in organic matter outpaces the stabilization of C generated from residues inside biogenic aggregates [15]. Soil biota dispersion is determined by microscale soil heterogeneity induced by changes in soil texture, resource distribution, and biotic interactions, which are all closely related to soil aggregation [16,17]. As a result, the development of soil aggregates serves as the foundation for the assembly and structuring of microbial communities, biotic interactions, and ecosystem function [18]. Continuous and abundant fertilizer in the absence of organic amendment N inputs frequently reduces earthworm populations and activity, whereas organic amendments have the reverse impact [19]. In this paper, FBO (earthworm biological organic fertilizer) was used as the organic fertilizer. Earthworms have the ability to change soil pH and organic matter content, reduce heavy metal enrichment by plants and improve the effective concentration of active components in soil [20]. Earthworm dung has the ability to improve soil water retention, increase soil porosity and reduce soil hardening. Additionally, earthworms support the regeneration of the soil microbial ecosystem and the creation of a healthy soil aggregate structure [21].

The evaluation of the effects of agricultural structures on soil quality, above- and below-ground biota and productivity is greatly aided by long-term agroecosystem trials [22,23]. At present, there are few studies on the application of earthworm bio-organic fertilizer to improve the soil fertility and tea quality of tea gardens. Furthermore, the mechanism of how earthworm bio-organic fertilizer promotes the formation of the soil mineral–organic complex is still unclear, and there are still few reports on the research of the soil mineral–organic complex based on synchrotron radiation-based Fourier transform-infrared microscopic imaging technology (SR-FTIR) [24]. More importantly, SR-FTIR has also been applied recently to characterize the distribution of organic carbon forms and clay minerals at the microscale level, with the advantages of high sensitivity and spatial resolution for obtaining in situ images [25,26].

We hypothesize that organic fertilization will stimulate nutrient transformation, sustain the yield production and promote the binding processes of organic and inorganic ligands in organic–mineral complexes compared with chemical fertilization alone, and thus may contribute to the accumulation of SOC in the tea plantation ecosystem. To test our hypothesis, a long-term field experiment on tea plantation with different fertilizers was conducted in a typical tea plantation region of China to investigate the effect of organic fer-

tilizer on soil quality and the mineral–organic complex. We analyzed data on soil chemical properties (soil total carbon, total nitrogen, dissolved organic carbon), soil structure (the organic–mineral complex, functional composition) and tea quality. More specifically, the effects of cultivating with and without fertilization on soil quality related to tea quality and the underlying mechanisms were explored. The results of this study will further elucidate the effects of soil nutrient deficiency on tea quality and help to comprehensively understand the internal mechanism of the tea garden soil response to different fertilization modes.

## 2. Materials and Methods

### 2.1. Study Site

The study was carried out in Yingde City, Guangdong Province (24°18′ N, 113°23′ E, approximately 0.4 hm²) (Figure 1). The area is characterized by a subtropical monsoon climate, with an average temperature of 21.1 °C and an average annual precipitation of 1906.2 mm [27]. The field experiment was set up in November 2009, with an experimental site of 80 m × 50 m (length × width); the row spacing between tea trees (variety "Jinxuan") was 1.5 m, a 2-m-wide isolation row was set up between each plot, and a 5-m-wide protection row was set up around the test field.

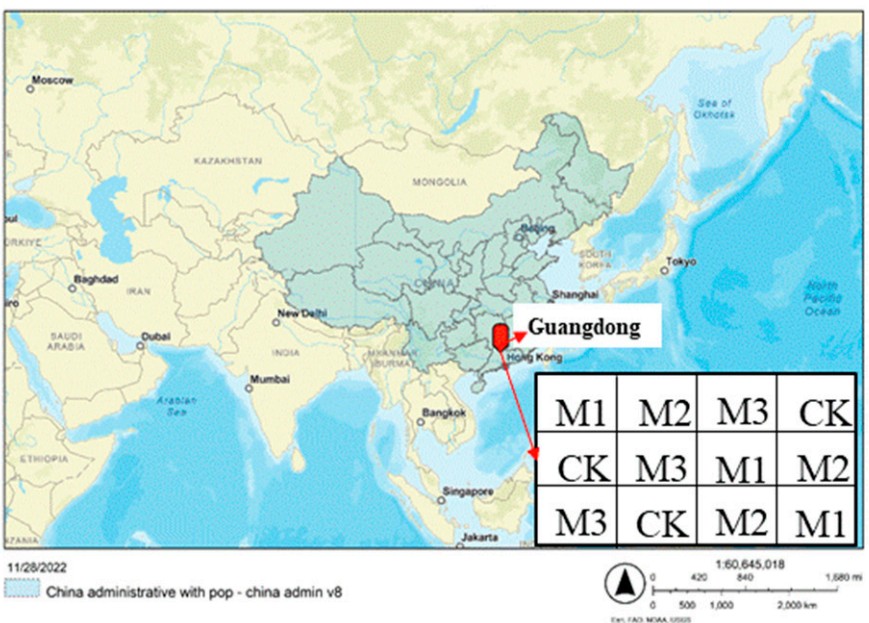

**Figure 1.** Location and distribution of the field treatments.

### 2.2. Experimental Design

The field treatments were divided into four groups: long-term 100% earthworm biological organic fertilizer (FBO) (M1), long-term 50% earthworm biological organic fertilizer plus 50% urea (M2), long-term urea (M3), and no-fertilizer control treatment (CK). Each treatment was repeated three times. In the first year, fertilizations were carried out according to the design dosage of the test plan. The organic fertilizers that were applied as the base fertilizers in a ditch were 100% FBO and 50% FBO, and the applied urea was top-dressed four times. In May 2010, earthworm inoculation was carried out in the plot treated with 100% FBO and 50% FBO according to the designed dosage (Table 1) [27]. Since then, in November of each year, only the prescribed dosage was used instead of earthworm inoculation. The tea garden soil was lateritic red soil, collected from the Yingde Test Base of the Institute of Drinking Plants, Guangdong Academy of Agricultural Sciences. The nutritional contents of the studied soil and fertilizers are shown in Table 2 [27].

**Table 1.** The quantity of the applied fertilizers for each treatment.

|  | Manure/kg | Straw/kg | Urea/kg | Earthworm/Individuals |
|---|---|---|---|---|
| M1 | 400 | 400 | - | 2000 |
| M2 | 200 | 200 | - | 1000 |
| M3 | - | - | 40 | - |
| CK | - | - | - | - |

**Table 2.** Nutritional content of the soil and fertilizers.

|  | Total P | Total N | Total K | Organic Matter (OM) | Organic Carbon (OC) | C/N |
|---|---|---|---|---|---|---|
| Soil | 0.39 g·kg$^{-1}$ | 0.99 g·kg$^{-1}$ | 24.86 g·kg$^{-1}$ | 17.66 g·kg$^{-1}$ | - |  |
| Manure | - | 10 g·kg$^{-1}$ | - | - | 180 g·kg$^{-1}$ | 18:1 |
| Straw | - | 8 g·kg$^{-1}$ | - | - | 410 g·kg$^{-1}$ | 51:1 |
| Urea | - | 46.4% | - | - | - | - |

The pot cultivation experiment was conducted to test the effect of fertilizers on tea quality and quantity in a glasshouse in Nanjing, Jiangsu Province, China. Jinxuan, an annual tea tree variety, was selected for the pot experiment. Each treatment was repeated six times. During cultivation, deionized water was regularly poured into the potted plants to maintain the normal growth of tea seedlings.

*2.3. Tea Sampling and Analyses*

When the new shoots of tea plants had a round of growth in the pot cultivation experiment, bud and leaf were harvested. The samples were stored in two parts, one in a clean aluminum box and dried in an oven, and the other fresh samples were stored for future use. The dry weight of one bud and one leaf was measured via the drying weighing method; the content of tea polyphenols was determined via folin phenol colorimetry [9] and the content of amino acids was determined via ninhydrin colorimetry [10].

*2.4. Soil Sampling*

Soil samples were collected in June 2018 and 2021. In each plot, surface soil (0–20 cm) samples were collected in five different positions using a 5-cm-diameter stainless steel soil sampling auger and then carefully mixed to form a composite. The soil was gently broken into small pieces which were 10 mm~12 mm in diameter along the natural structural plane. The coarse roots, stones and litter residues was discarded. About 1 kg of soil samples were collected in each plot and stored in the closed wooden box, and then taken to the laboratory. All soil samples were kept according to two categories: the first was to be air-dried and sieved with 20-mesh and 100-mesh sieves for analysis, and the other was to be placed at 4 °C in the refrigerator for further use.

*2.5. Analyses of Soil's Chemical Properties*

The total carbon and nitrogen were determined using a vario MACRO cube series element analyzer (Elementar Analysensysteme GmbH, Germany). Dissolved organic C (DOC) in soil was extracted with deionized water in a 1:5 (*v/v*) soil-to-water ratio by shaking for 30 min, followed by centrifugation for 15 min (7570× *g*) and filtration (<0.45 μm), before being analyzed using a combustion oxidation nondispersive infrared absorption method with a Vario TOC Cube Total Organic Carbon Analyzer (Elementar Trading Co., Ltd., Shanghai, China).

*2.6. Analyses of Soil Enzyme Activity*

The activity of urease was quantified by mixing amended soil (5 g of dry soil) with 10 mL urea solution (10%), 1 mL methylbenzene and 20 mL citrate buffer (pH 6.7), followed by incubation at 37 °C for 24 h. The indophenol colorimetric method was used to quantify the resulting $NH_4^+$ from urease-mediated urea hydrolysis, and calorimetric measurements were determined at 578 nm, with the enzymatic activity described using mg $NH_4^+$-N $kg^{-1}$ $h^{-1}$. To determine the activities of nitrate reductase, nitrite reductase, 0.5% $NaNO_2$ solution, 0.5% $NH_2OH$ and 1.0 mL volumes of 1% $KNO_3$ solution, 0.5% $NaNO_2$ solution and 0.5% $NH_2OH$ were added to 1.0 g soil samples, respectively. The mixtures were incubated at 30 °C for 24 h, and the consumption rates of $NO_3^-$-N and $NO_2^-$-N were measured to determine the activities of nitrate reductase and nitrite reductase [28,29].

*2.7. Analyses of Soil Enzyme Activity*

We selected the particles from all of the fresh soil treatments and then froze them at −20 °C, embedded them in deionized water and then sectioned and transferred them to low-E slides (Kevley Technologies, Chesterland, OH, USA). The slides were then preserved in a desiccator until they could be analyzed. The distribution of organic and inorganic groups was evaluated using SR-FTIR spectroscopy at the National Center for Protein Science Shanghai (NCPSS) and Shanghai Synchrotron Radiation Facility's beamline BL01B1 (SSRF). A Thermo Nicolet 6700 FTIR spectrometer in reflectance mode was used to record the spectra with the following settings: $20 \times 20$ $mm^2$ aperture size, $10 \times 10$ $mm^2$ step size, 2 $cm^{-1}$ resolution, spectral range 4000–650 $cm^{-1}$ and 64 scans.

*2.8. Statistical Analysis*

We processed SR-FTIR spectroscopy using Omnic software Version 9.0 (Thermo Fisher Scientific Inc, Waltham, MA, USA). Then, the functional groups were analyzed using Origin 9.0 software. SPSS software Version 18.0 for Windows was used for analyzing the data (means ± SD, $n = 3$ for soil and $n = 6$ for tea samples) using ANOVA. We used Duncan's multiple range test at $p \leq 0.05$ to test the differences between different treatments. Pearson's correlation coefficient (r) values were performed to examine linear correlations at $p < 0.05$.

## 3. Results

*3.1. Dry Weight and Biochemical Results of One Bud and One Leaf of Tea Seedlings across Different Treatments*

The growth and biochemical analysis results of tea seedlings are shown in Table 3. After fertilization, there was no significant difference in the dry weight of one bud and one leaf among the treatments, but the continuous input of organic amendments to the soils relatively increased the amount of tea yield compared with no fertilization and chemical fertilization. Similarly, the difference in tea polyphenol content among the treatments reached a significant level, and the content of tea polyphenol in the treatment of adding FBO fertilizer was significantly higher than that in the treatment of CK and M3. Moreover, amino acid content was larger in those of organic fertilized treatments (i.e., M1 and M2) than those of chemical fertilized treatment (M3) and no fertilized treatment (CK).

**Table 3.** Growth and biochemical characteristics of tea seedlings under different treatments.

| Treatments | The Dry Weight of One Bud and One Leaf (g) | Tea Polyphenols ($\mu g \cdot mL^{-1}$) | Amino Acids ($mg \cdot mL^{-1}$) |
|---|---|---|---|
| CK | 0.44 ± 0.16 a | 1.11 ± 0.04 b | 1.15 ± 0.03 b |
| M1 | 0.71 ± 0.11 a | 2.26 ± 0.07 a | 2.78 ± 0.33 ab |
| M2 | 0.83 ± 0.14 a | 2.75 ± 0.23 a | 3.54 ± 0.85 a |
| M3 | 0.68 ± 0.15 a | 1.29 ± 0.07 b | 1.25 ± 0.08 b |

Note: fertilizer treatment: CK, control without fertilizer; M1, 100% earthworm biological organic fertilizer; M2, 50% earthworm biological organic fertilizer plus 50% urea; M3, 100% chemical fertilizer. Data are presented as means and standard errors ($n = 6$). Different letters following data in the same row indicate significant differences at $p < 0.05$.

### 3.2. Total Carbon and Nitrogen in Soil

There were significant differences in soil nutrients across the different fertilization treatments. The content of total nitrogen and total carbon in tea garden soil of different years is shown in Figure 2. Soil total nitrogen and carbon content in 2021 were higher than that in 2018. In 2018, the total nitrogen and total carbon content of soil fertilization treatment was significantly higher than that of no fertilization treatment (CK). With increased time, the total nitrogen and total carbon content of the soil of the soil fertilization treatment was higher than that of the no fertilization treatment in 2021. Notably, long-term organic and inorganic fertilization treatments have been shown to significantly increase the concentration of carbon and nitrogen in soils when compared to inorganic fertilization and no fertilization treatments.

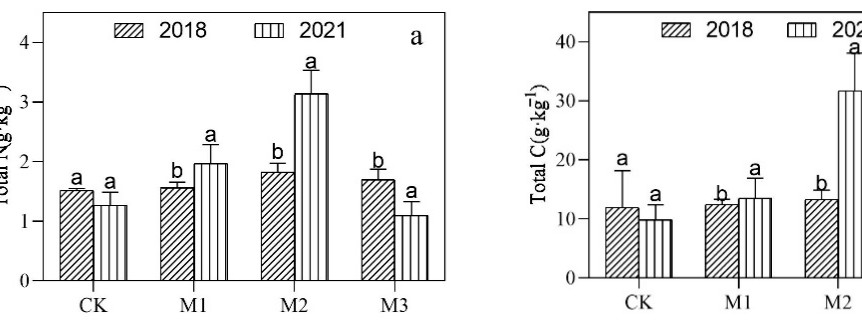

**Figure 2.** Soils' total carbon and nitrogen contents. Notes: Fertilizer treatment as described in Table 3. Values within the same column followed by different letters are significantly different at *p* < 0.05.

### 3.3. Dissolved Organic Carbon

Both organic fertilized treatments (i.e., M1 and M2) increased the dissolved organic carbon (DOC) contents in tea garden soil (Figure 3). In 2018, the content of dissolved organic carbon in the fertilization treatment was slightly higher than that in the non-fertilization treatment, and the content of DOC in the organic–inorganic combined fertilization treatment was the highest. Different from 2018, the content of DOC in 100% FBO treatment was significantly higher than that in the no-fertilizer treatment in 2021. In addition, the DOC contents obviously increased in organic fertilized treatments from 2018 to 2021 while the DOC contents decreased in chemical fertilized treatment (M3) and no fertilized treatment (CK).

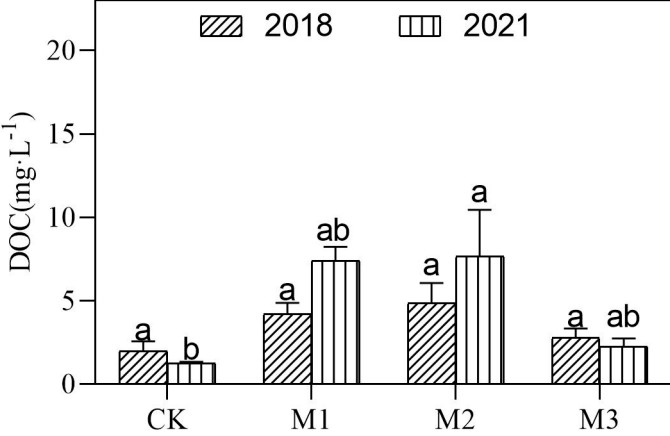

**Figure 3.** Soils' dissolved organic carbon (DOC) content under different fertilization treatments. Notes: Fertilizer treatment as described in Table 3. Values within the same column followed by different letters are significantly different at *p* < 0.05.

### 3.4. Soil Enzyme Activity

As shown in Table 4, the effects of fertilization on soil enzyme activities in tea plantation differed significantly with the different fertilizer treatments. In the 0–20 cm soil layer in 2021, the activity of the three enzymes in the fertilization treatment was higher than that in the non-fertilization treatment, and the difference with CK reached a significant level. Among them, the urease activity was the highest in the 100% FBO treatment, and the nitrate reductase and nitrite reductase activity were the highest in the organic–inorganic combination treatment.

**Table 4.** Effects of different fertilization treatments on soil enzyme activities in tea plantation.

| Treatments | Urease ($g^{-1} \cdot kg^{-1} \cdot h^{-1}$) | Nitrate Reductase ($mg \cdot g^{-1} \cdot h^{-1}$) | Nitrite Reductase ($mg \cdot g^{-1} \cdot h^{-1}$) |
|---|---|---|---|
| CK | $97.58 \pm 28.16$ d | $0.0001 \pm 0.0001$ b | $0.006 \pm 0.001$ b |
| M1 | $217.32 \pm 38.45$ a | $0.0004 \pm 0.0002$ b | $0.008 \pm 0.001$ b |
| M2 | $145.04 \pm 11.71$ b | $0.0037 \pm 0.001$ a | $0.026 \pm 0.002$ a |
| M3 | $106.97 \pm 11.26$ c | $0.0003 \pm 0.0002$ b | $0.007 \pm 0.001$ b |

Note: fertilizer treatment as described in Table 3. Data are presented as means and standard errors ($n = 3$). Different letters following data in the same row indicate significant differences at $p < 0.05$.

### 3.5. Synchrotron Radiation Infrared Microscopic Imaging and Microarea Infrared Spectrum Characteristics of Soil Organo-Mineral Complex

The absorption characteristic peak of the infrared spectrum and its attribution are shown in Table 5. According to the infrared spectra in Figures 4 and 5, the characteristic peaks ($3620$ cm$^{-1}$) of clay minerals are from scratch, and the characteristic peaks of macromolecular organic substances are either from scratch (such as $1650$ cm$^{-1}$ and $2920$ cm$^{-1}$) or gradually increase in intensity ($1080$ cm$^{-1}$). The colors of functional groups in these photographs ranged from red to blue, according to the relatively strong SR-FTIR absorbance to the comparatively weak one [8,23,30].

**Table 5.** Infrared spectra absorption peak bands and assignments.

| Infrared Spectra Absorption Peak Position (cm$^{-1}$) | Absorption Peak Assignment |
|---|---|
| 3620 | Clay minerals (alcohol hydroxyl-OH stretching vibration) |
| 3150–2920 | Aliphatic group (C-H extension of $CH_2$) |
| 1650–1530 | Protein (aromatic C=C stretching vibration) |
| 1170–950 | Polysaccharides (C-OH stretching) |
| 1200–970 | Si-O-Si vibration |
| 950–900 | Al-O vibration |
| 695 | Fe-O vibration |

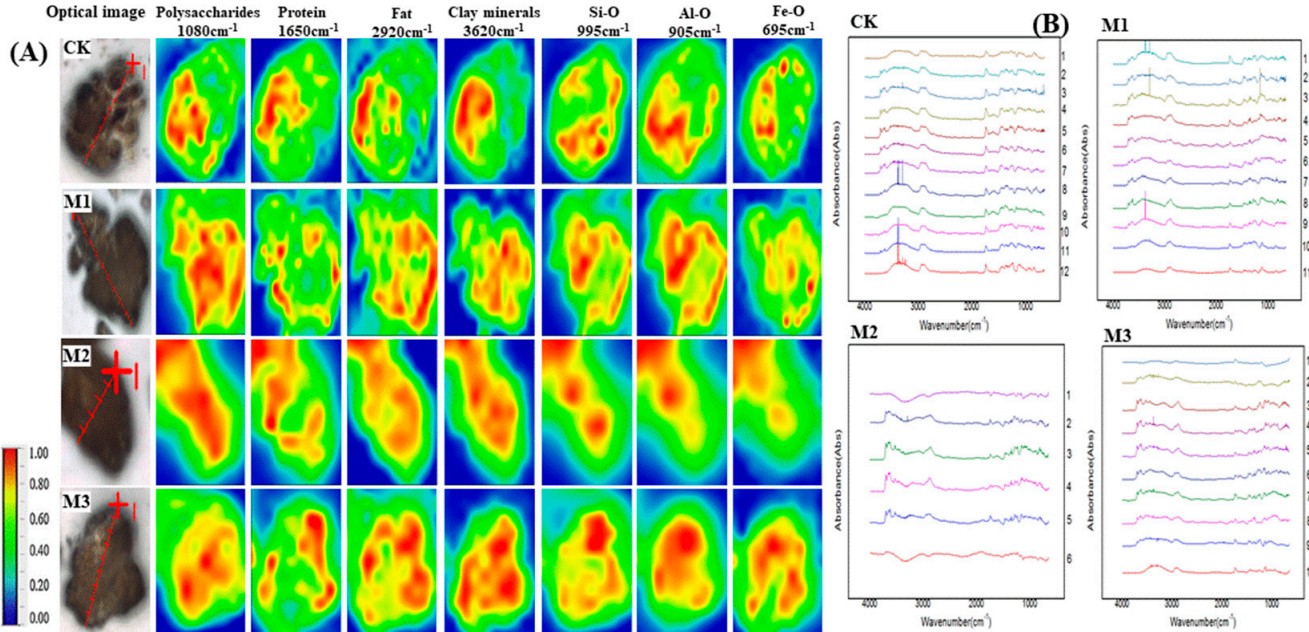

**Figure 4.** Synchrotron radiation infrared microimaging image and microregional infrared spectrum of soil organo-mineral complex under different fertilization treatments in 2018. (**A**) Distribution maps of functional groups in tea plantation soil detected by SR-based FTIR strategy. Note: Red represents high intensity and blue represents low intensity. (**B**) 1D SRFTIR spectra extracted from regions of interest (ROIs) which distributed on the red line in (**A**). Fertilizer treatment as described in Table 3.

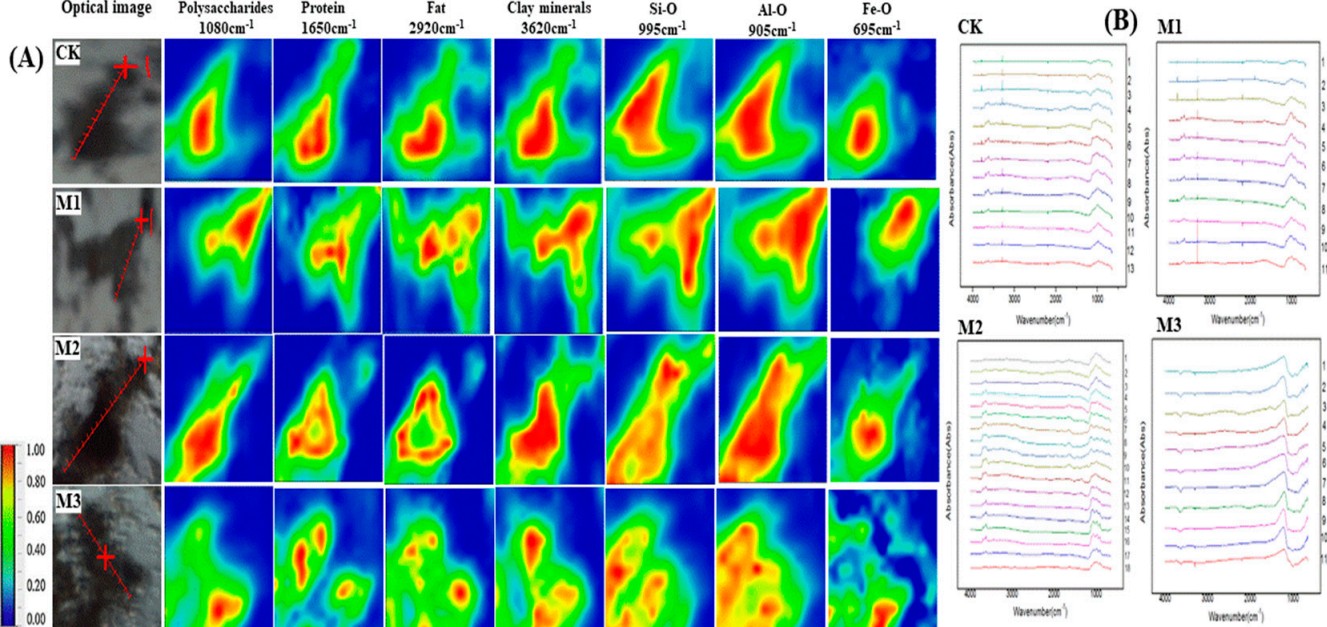

**Figure 5.** Synchrotron radiation infrared microimaging image and microregional infrared spectrum of soil organo-mineral complex under different fertilization treatments in 2021. (**A**) Distribution maps of functional groups in tea plantation soil detected by SR-based FTIR strategy. Note: Red represents high intensity and blue represents low intensity. (**B**) 1D SRFTIR spectra extracted from regions of interest (ROIs) which distributed on the red line in (**A**). Fertilizer treatment as described in Table 3.

In 2018 and 2021, the SR-FTIR spectromicroscopy also revealed the distribution pattern of distinct groups within the soil particles treated with 100% FBO (M1), 50% FBO and 50% urea (M2), urea (M3) and no fertilizer (CK) (Figures 4 and 5). In 2018, the distribution of inorganic and macromolecular organic matter in soil treated by CK and M1 was relatively dispersed, while that of M2 and M3 was relatively concentrated. Similarly, in the same treatment, the distribution laws of inorganic substances and macromolecular organic substances were basically the same. In 2021, the distribution of clay minerals, silicon, aluminum, iron and other inorganic substances and macromolecular organic matters (protein, fat, polysaccharide) in soil had a high heterogeneity, especially in M2 and M3 treatments. Different kinds of functional groups have different distribution laws in different soil environments. The distribution of clay minerals and iron oxides is similar to that of polysaccharides, but different from that of proteins and fats; the distribution of protein and fat, and silicon and aluminum oxides is similar. The dispersion degree of macromolecular organics in soil microaggregates treated by M1, M2 and M3 is higher than that of CK. The macromolecular organics and clay minerals in soil microaggregates treated by CK mainly exist as large particles. The results show that clay minerals are mainly concentrated in the micro aggregates, and the distribution of macromolecular organic matter has strong heterogeneity. This conclusion provides strong evidence for infrared microscopic imaging of soil microaggregates.

### 3.6. Correlation Analysis of Inorganic Matter and Macromolecular Organic Matter in Soil Organo-Mineral Complex

The correlation analysis results of minerals (clay minerals, silicon/aluminum/iron oxides) in tea garden soil microaggregates and macromolecular organic functional groups are shown in Table 6. The results show that the order of the determination coefficients ($R^2$) of clay minerals and organic functional groups in the fertilization treatment was aliphatic–clay minerals > polysaccharide–clay minerals > protein–clay minerals, that is, the affinity of clay minerals in soil microaggregates was the highest with aliphatic groups, followed by polysaccharide and the lowest with protein. The tea garden soil samples without fertilization showed a $R^2$ order different from that of fertilization treatment: aliphatic group–clay minerals > protein–clay minerals > polysaccharide–clay minerals. The determination coefficients ($R^2$) of iron/aluminum/silicon oxides and organic functional groups in the non-fertilization and FBO treatments were iron/aluminum/silicon oxides–polysaccharide > iron/aluminum/silicon oxides–protein > iron/aluminum/silicon oxides–aliphatic group, that is, iron/aluminum/silicon oxides had the highest affinity with polysaccharides, followed by protein, and the lowest affinity with the aliphatic group. However, the treatment of single fertilizer application showed a different $R^2$ order of iron/aluminum/silicon oxide–aliphatic group > iron/aluminum/silicon oxide–polysaccharide > iron/aluminum/silicon oxide–protein.

**Table 6.** The correlation analysis between organic functional groups and minerals in soil aggregates as affected by different fertilizer treatments using SR-FTIR in 2018 and 2021.

| Treatment | Year | Clay Minerals | | | Si-O | | | Al-O | | | Fe-O | | |
|---|---|---|---|---|---|---|---|---|---|---|---|---|---|
| | | Polysaccharides | Protein | Fat | Polysaccharides | Protein | Fat | Polysaccharides | Protein | Fat | Polysaccharides | Protein | Fat |
| CK | 2018 | $Y = 0.378x + 1.289$ $R^2 = 0.758$ | $Y = 0.611x + 0.162$ $R^2 = 0.833$ | $Y = 0.498x + 1.296$ $R^2 = 0.788$ | $Y = 0.525x + 1.052$ $R^2 = 0.724$ | $Y = 0.766x - 0.133$ $R^2 = 0.646$ | $Y = 0.551x + 1.134$ $R^2 = 0.475$ | $Y = 0.668x + 0.888$ $R^2 = 0.685$ | $Y = 1.071x - 0.475$ $R^2 = 0.738$ | $Y = 0.827x + 0.825$ $R^2 = 0.628$ | $Y = 0.629x + 1.095$ $R^2 = 0.729$ | $Y = 0.947x - 0.094$ $R^2 = 0.693$ | $Y = 0.670x + 1.170$ $R^2 = 0.493$ |
| | 2021 | $Y = 0.656x - 0.041$ $R^2 = 0.780$ | $Y=0.595x - 0.055$ $R^2 = 0.834$ | $Y = 0.737x + 0.017$ $R^2 = 0.931$ | $Y = 0.682x - 0.098$ $R^2 = 0.872$ | $Y = 0.531x - 0.055$ $R^2 = 0.684$ | $Y = 0.605x + 0.048$ $R^2 = 0.645$ | $Y = 0.922x - 0.100$ $R^2 = 0.918$ | $Y = 0.740x - 0.067$ $R^2 = 0.767$ | $Y = 0.827x + 0.041$ $R^2 = 0.695$ | $Y = 1.828x + 0.029$ $R^2 = 0.844$ | $Y = 1.515x + 0.030$ $R^2 = 0.751$ | $Y = 1.561x + 0.168$ $R^2 = 0.579$ |
| M1 | 2018 | $Y = 0.493x + 1.047$ $R^2 = 0.662$ | $Y = 0.334x + 0.208$ $R^2 = 0.642$ | $Y = 0.631x + 0.835$ $R^2 = 0.713$ | $Y = 0.772x + 0.710$ $R^2 = 0.817$ | $Y = 0.454x + 0.050$ $R^2 = 0.598$ | $Y=0.897x + 0.498$ $R^2 = 0.725$ | $Y = 0.790x + 0.685$ $R^2 = 0.825$ | $Y = 0.465x + 0.035$ $R^2 = 0.604$ | $Y = 0.916x + 0.470$ $R^2 = 0.729$ | $Y = 0.845x + 0.746$ $R^2 = 0.700$ | $Y = 0.510x + 0.060$ $R^2 = 0.538$ | $Y = 1.056x + 0.472$ $R^2 = 0.719$ |
| | 2021 | $Y = 0.234x + 0.118$ $R^2 = 0.096$ | $Y = 0.149x + 0.151$ $R^2 = 0.060$ | $Y = 0.652x + 0.038$ $R^2 = 0.897$ | $Y = 0.657x - 0.104$ $R^2 = 0.798$ | $Y = 0.462x - 0.012$ $R^2 = 0.620$ | $Y = 0.163x + 0.282$ $R^2 = 0.053$ | $Y = 0.282x + 0.114$ $R^2 = 0.107$ | $Y = 0.191x + 0.144$ $R^2 = 0.077$ | $Y = 0.681x + 0.072$ $R^2 = 0.754$ | $Y = 1.586x + 0.087$ $R^2 = 0.779$ | $Y = 1.118x + 0.121$ $R^2 = 0.608$ | $Y = 0.303x + 0.338$ $R^2 = 0.029$ |
| M2 | 2018 | $Y = 0.634x + 0.305$ $R^2 = 0.864$ | $Y = 0.444x + 0.282$ $R^2 = 0.816$ | $Y = 0.831x - 0.107$ $R^2 = 0.961$ | $Y = 1.138x - 0.045$ $R^2 = 0.954$ | $Y = 0.763x + 0.071$ $R^2 = 0.822$ | $Y = 1.330x - 0.407$ $R^2 = 0.839$ | $Y = 1.324x - 0.045$ $R^2 = 0.938$ | $Y = 0.895x + 0.064$ $R^2 = 0.824$ | $Y = 1.530x - 0.394$ $R^2 = 0.808$ | $Y = 1.642x + 0.220$ $R^2 = 0.857$ | $Y = 1.124x + 0.237$ $R^2 = 0.770$ | $Y = 1.823x - 0.049$ $R^2 = 0.680$ |
| | 2021 | $Y = 0.834x - 0.061$ $R^2 = 0.848$ | $Y = 0.544x - 0.028$ $R^2 = 0.768$ | $Y = 0.372x - 0.023$ $R^2 = 0.621$ | $Y = 0.979x - 0.106$ $R^2 = 0.766$ | $Y = 0.501x - 0.046$ $R^2 = 0.643$ | $Y = 0.345x - 0.036$ $R^2 = 0.530$ | $Y = 0.163x - 0.117$ $R^2 = 0.835$ | $Y = 0.685x - 0.061$ $R^2 = 0.739$ | $Y = 0.483x - 0.052$ $R^2 = 0.636$ | $Y = 1.693x + 0.045$ $R^2 = 0.814$ | $Y=1.104x + 0.041$ $R^2 = 0.739$ | $Y = 0.801x + 0.015$ $R^2 = 0.671$ |
| M3 | 2018 | $Y = 0.517x + 0.431$ $R^2 = 0.437$ | $Y=0.364x + 0.250$ $R^2 = 0.661$ | $Y = 0.646x + 0.503$ $R^2 = 0.730$ | $Y = 1.054x + 0.047$ $R^2 = 0.745$ | $Y = 0.544x + 0.136$ $R^2 = 0.598$ | $Y = 1.034x + 0.247$ $R^2 = 0.759$ | $Y = 0.838x + 0.247$ $R^2 = 0.630$ | $Y = 0.466x + 0.214$ $R^2 = 0.589$ | $Y = 0.876x + 0.403$ $R^2 = 0.730$ | $Y = 0.808x + 0.311$ $R^2 = 0.345$ | $Y = 0.625x + 0.127$ $R^2 = 0.632$ | $Y = 1.052x + 0.325$ $R^2 = 0.626$ |
| | 2021 | $Y = 0.867x - 0.017$ $R^2 = 0.824$ | $Y=0.267x + 0.035$ $R^2 = 0.478$ | $Y = 0.485x + 0.029$ $R^2 = 0.832$ | $Y = 0.555x + 0.011$ $R^2 = 0.653$ | $Y = 0.178x + 0.039$ $R^2 = 0.411$ | $Y = 0.315x + 0.041$ $R^2 = 0.677$ | $Y = 0.692x - 0.029$ $R^2 = 0.755$ | $Y = 0.217x + 0.028$ $R^2 = 0.455$ | $Y = 0.376x + 0.028$ $R^2 = 0.719$ | $Y = 2.967x + 0.141$ $R^2 = 0.549$ | $Y = 0.390x + 0.127$ $R^2 = 0.055$ | $Y = 0.943x + 0.176$ $R^2 = 0.176$ |

## 4. Discussion

### 4.1. Effects of Different Fertilization Treatments on the Growth of Tea Seedlings

Previous findings about the effects of earthworms on plant development have generated controversy, most likely as a result of the wide variety of conditions (such as earthworm species, plant types or soil qualities) under which the research was conducted [31]. Van Groenigen et al. [32] showed the earthworm enhancement of plant manufacturing which may want to be attributed to the enhancement of nutrient mineralization and soil shape indicated by using tomato and spinach yields. In addition, they additionally illustrated that earthworms produced a quarter more crop yield across a vast variety of situations, with the extent of the expansion relying on the kind of fertilizer and the primary soil properties. Ayeni et al. [33] found that, compared with the control, organic fertilizer (OM) and organo-mineral fertilizer (OMF) significantly increased ($p < 0.05$) maize plant height, number of leaves, leaf area, stover yield, root dry matter and grain yield. Results from Giannakis et al. showed that biosolid addition enhanced plant growth, fresh weight, root weight, stem height and leaf number of tomato plants [34]. This conclusion agrees well with the results of the current field study, considering that the current experimental factors included continuous FBO amendment. Compared with the non-fertilization treatment, the fertilization treatment has a significant impact on the dry weight of a bud and a leaf, and the tea polyphenol and amino acid content of the tea seedlings in a round of growth. The two groups of treatment with FBO have a significant difference in the tea polyphenol and amino acid content compared with the non-fertilization treatment and the single-chemical fertilizer treatment. It is proved that the addition of FBO, especially the combination of FBO and inorganic fertilizer, can improve soil structure and tea quality.

### 4.2. Effects of Different Fertilization Treatments on Soil Carbon and Nitrogen

Soil lively natural carbon (AOC) is an especially environmentally friendly natural carbon in soil, which is handy to be degraded and mineralized via soil microorganisms. Its ecological and environmental outcomes are by and large mirrored in its use of soil nutrients. Soil lively natural carbon (AOC) is a foremost indicator for evaluating soil fertility, which is affected with the aid of soil tillage and fertilization administration [35]. Yang et al. concluded that grassland restoration increased soil C sequestration primarily by microbial necromass (mainly bacterial necromass), and is affected by abiotic and biotic factors, as well as plant C input [36]. Sun M et al. [37] and Gao J et al. [38] have found that both the single application of organic fertilizer and organic matter mulching are conducive to an increase in DOC content and soil nutrient cycling. The higher DOC content in soil treated with M1 and M2 indicated that the improved soil treated with earthworm biological organic fertilizer technology was suitable for tea planting. Mikha et al. found that the long-term improvement of mineral (NPK) fertilizer utility may additionally minimize the soil's aggregate balance and consequently improve the safety of soil's natural carbon in microaggregates [39], since the long-term application of extra mineral fertilizer and/or pesticides probably counteracts the build-up of carbon-wealthy soils. This observation supports Jenkinson et al.'s earlier findings that mineral N addition will increase the decomposition price of natural residues through pleasurable N necessities of microorganisms [40]. In addition, the overall fertility of the tea garden soil in the Guangdong tea area was low, and more organic fertilizer was needed. The results of this research showed that 100% FBO treatment and 50% FBO treatment had the same change rule of total carbon, soluble organic carbon and total nitrogen content, which were significantly higher than that of no-fertilization and single-fertilization treatment, and the content increased with an increase in fertilization years. The nutrient content released was constrained, the rate of breakdown and transformation of soil organisms into organic matter was slow and the nutrient content released at later stages of the intake of organic materials accumulated and rose. Wang SQ et al. [41] also showed that after applying organic fertilizer, the content of total carbon and total nitrogen in the soil was significantly higher than that without fertilization, which supports our results.

*4.3. Effects of Different Fertilization Treatments on Soil Enzyme Activities*

Yang et al. suggest that it is essential to conduct long-term, multiple-factor experiments to assess the response of soil microbial diversity to global change because multiple global change factors often occur simultaneously [42]. Involved in numerous significant biochemical processes in soil and closely related to soil fertility, soil enzymes are a type of unique protein with biochemical and catalytic properties [43]. Liang et al. found that soil amendments (such as compost, farm yard manure, and municipal solid waste) considerably boost the enzymatic activity of urease, glucosidase, alkaline phosphatase and o-diphenyl oxidase, when compared to soil treated with chemical fertilizers [44]. Lv et al. achieved the results that herb residue vermicompost supported greater enzyme activities than conventional NPK fertilizer [45]. The effect of fertilization on soil enzyme activity was statistically significant ($p < 0.05$) in this study. Long-term use of organic fertilizer can increase soil fertility by accelerating soil nutrient activation, increasing soil nutrient content and maintaining the balance of accessible nutrients [46]. Additionally, organic fertilizer can improve the activity of enzymes and microorganisms involved in nutrient transformation, increasing the amount of nutrients accessible in the soil [47]. Our research results showed that the activities of urease, nitrate reductase and nitrite reductase in the treatment with FBO were significantly higher than those in the treatment without fertilization and the treatment with chemical fertilizer alone. Similar results were also obtained from the study by Tang JC et al. [27], who conducted experiments on earthworm bio-organic fertilizer in tea gardens for five consecutive years, proving that the 100% FBO treatment significantly improved the activities of several enzymes closely related to soil nutrient transformation, further proving the improvement effect of earthworm bio-organic fertilizer technology on soil.

*4.4. Effects of Different Fertilization Treatments on Soil Organo-Mineral Complex*

Soil structure is the main factor affecting soil quality. The two physical forms of soil organic matter are aggregates and particles of various sizes suspended in the aqueous phase, or individual molecules completely encapsulated in water molecules [19]. An appropriate soil structure can optimize the nutritional function of crops and improve the delivery of nutrients. It can also control water, heat, air, biology and other environmental factors. Mineralogy can play a role in the preservation of specific functional groups. Adomako et al. [20] showed that iron and aluminum oxides are secondary components of soil's clay minerals, however because of their high activity, they are quickly harmed by the environment. Qian W et al. [48] considered that iron oxides have a large specific surface area, which contributes to the creation and stability of small-particle-size aggregates, making the interaction between organic matter and metal oxides more crucial for the formation of aggregates. Coordinated minerals, iron oxides and aluminum oxides can help to restore together organic matter and soil's organic–inorganic complex, preserving organic matter. ChaplotV et al. [49] showed that the stability of polymer water is closely related to soil structure. The stability of the soil structure improves with polymer water content, which is beneficial for increasing soil fertility. This is similar to the test results. The correlation analysis results of organic mineral functional groups of soil aggregates show that the combination of organic functional groups with iron and aluminum oxides is good, and the infrared microscopic image in 2021 shows the obvious protection of hydrophilic organic matter (fat), which affects the distribution and stability of soil aggregates to a certain extent. The stability of the polymer is the reflection of soil properties such as water holding capacity, permeability and anti-scour ability, and is an important index for evaluating soil stability.

Long-term organic or inorganic amendments directly or indirectly induce changes in soil's physiochemical and biological properties. The application of organic fertilizer revealed the ability of soil aggregates to fix soil's organic carbon through mineral–organic complexes. The protection of clay minerals and iron/aluminum oxides toward hydrophilic compounds (fats) reduced the possibility of being decomposed by microorganisms and extracellular enzymes. Our study gave evidence to support the notion that organo-mineral

interactions are bound by Al−OH, Fe−OH and Si-O vibrations in soil microaggregates, which should be considered as important contributions to long-term C storage [8]. Organic matter adsorbed on clay minerals and iron/aluminum oxides was more stable. Therefore, it is necessary to find organic fertilizer that can promote soil particle aggregation, has strong stability and improves soil structure and soil fertility.

## 5. Conclusions

Over a 12-year period in a cultivated tea field experiment, the yield and quality of tea were found to be closely related to the soil fertility, and the fertile tea garden soil was the basic guarantee for achieving high quality and a high yield of the tea garden. The tea garden soil in Guangdong Province is mainly laterit soil or red soil with strong acidity. This study established that the application of FBO fertilizer can not only improve the quality and quantity of tea, but also relieve the soil degradation caused by the application of chemical fertilizer and nutrient imbalance in the tea area of Guangdong. The application of organic and inorganic fertilizer (M2) was more effective than organic fertilizer (M1) in improving the contents of total carbon and nitrogen in the soils. The higher DOC content in soil treated with M1 and M2 indicated that the improved soil treated with earthworm biological organic fertilizer technology was suitable for tea planting. Our results indicated that soil enzyme activities varied under different fertilization regimes which depended on enzyme types. Additionally, the continuous bio-organic fertilizer experiment in the tea garden has proved that the replacement of chemical fertilizer with organic fertilizer can improve the ability of clay minerals and iron/aluminum/silicon oxides to protect aliphatic groups, polysaccharides and proteins, which is conducive to the repair of soil structure in the tea garden and an overall improvement in soil fertility.

**Author Contributions:** Conceptualization, methodology, formal analysis, writing—original draft preparation, project administration, H.L.; investigation, formal analysis, writing—original draft preparation, Y.Z. and H.M.; resources, J.L.; project administration, X.C. and Q.H.; conceptualization, writing—review and editing, supervision, funding acquisition, X.L. and J.T. All authors have read and agreed to the published version of the manuscript.

**Funding:** Supported by key research and development program of Sichuan Province (23ZDYF3044), expert workstation of Yunnan Province (202105AF150045), national key research and development program of China (2022YFD1601800), national natural science foundation of China (32172628, 31972457) and China agriculture research system of MOF and MARA (CARS-19).

**Institutional Review Board Statement:** Not applicable.

**Informed Consent Statement:** Not applicable.

**Data Availability Statement:** Not applicable.

**Acknowledgments:** We thank the staff from the BL01B beamline of the National Facility for Protein Science in Shanghai (NFPS) at the Shanghai Synchrotron Radiation Facility for assistance during data collection.

**Conflicts of Interest:** The authors declare no conflict of interest.

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
