# Peer review of "Effects of Long-Term Application of Earthworm Bio-Organic Fertilization Technology on Soil Quality and Organo-Mineral Complex in Tea Garden"

_forests, doi:10.3390/f14020225_

Round 1

Reviewer 1 Report

The authors have provided an article type manuscript focused on Effects of Long-Term Application of Earthworm Bio-organic Fertilization Technology on Soil Quality and organo-mineral complex in Tea Garden. Please see my suggestions bellow:

L33. 3.2 million hm2 or km2? Please check and revise if necessary.

L42. SOC abbreviation. As the Instructions for authors require, Abbreviations should be defined the first time they appear in each of 3 sections: the abstract; the main text; under the first figure or table. When defined for the first time, the acronym/abbreviation/initialism should be added in parentheses after the written-out form. Please revise the entire manuscript in this regard.

Paragraph/phrase L40-42 must be better developed. “It is well known that the addition of organic fertilizer has beneficial effects on soil quality, because it improves the soil porosity, it increases the SOC content [4], it modifies the soil enzymes/ enzymology (I suggest here checking and referring to  https://doi.org/10.37358/RC.18.10.6590   and https://doi.org/10.37358/RC.17.10.5864 ) and it contains all essential nutrients which are slowly released [5].

There is nothing about climate changes influences on the fertilisation, which is one of the main factors in the soil management. Please complete after checking https://doi.org/10.1007/s11356-021-14127-7

As the topic is not a new one, In the last paragraph of Introduction, please highlight better the special aspects that your study brings to the field, and what differentiate this paper from other in the same topic.

L101. Please provide a satellite image of the area in the study.

L104. Text in the parenthesis seems unfinished. Please check

For all the apparatus used in the experimental part, please provide / complete 4 information, as follows: model, producer/manufacturer, city, and country.

L150. The section must be named as Results (remove Analysis), as the Instructions for Authors requests.

L178. After the title of the figure, please explain the meaning of the letters a, b used on the figures. Also, all abbrev. used must be explained, as I mentioned above.

Some figures are blurred. Please, if you can, to provide better/best quality figures.

L352 and L 356, both phrases beginning with “Therefore”. I suggest English revising avoiding repetition.

L358. Please add and detail the strengths and the weakness of your results/study.

Reviewer 2 Report

Overall, manuscript contains useful data and can be a good contribution to literature, However, the presentation of manuscript and the data is not ideal and needs thorough revision.
The write-up of manuscript is very general and  Authors need to present their data in a concise and comprehensive way. No need to explain again and again the same fact under different headings or by different techniques. The output may be explained under one section and the related techniques/data may be used to justify the observed output. Authors also need to shorten the length of paragraphs. There are several paragraphs, especially in introduction section, which are too long. One paragraph need to present one message in a clear and concise

The objective of current study is to clear to readers. In this manuscript, authors have used numerous abbreviations. The use of abbreviations does not follow underlying rules. Abbreviations must be described at their first appearance in the text. Please verify the use of abbreviations.  I found some citations are very old like 1985, 1998, 2002, so kindly replace all those citation with the below mentioned citation which are most recent and updated. i will recheck the revised version with suggestions inclusion 

1.         Gong, X., Wang, L., Mou, Y., Wang, H., Wei, X., Zheng, W.,... Yin, L. (2022). Improved Four-channel PBTDPA Control Strategy Using Force Feedback Bilateral Teleoperation System. International Journal of Control, 20(3), 1002-1017. doi: 10.1007/s12555-021-0096-y

2.         Zheng, W., Tian, X., Yang, B., Liu, S., Ding, Y., Tian, J.,... Yin, L. (2022). A Few Shot Classification Methods Based on Multiscale Relational Networks. Applied Sciences, 12(8). doi: 10.3390/app12084059

3.         Ban, Y., Liu, M., Wu, P., Yang, B., Liu, S., Yin, L.,... Zheng, W. (2022). Depth Estimation Method for Monocular Camera Defocus Images in Microscopic Scenes. Electronics (Basel), 11(13), 2012. doi: 10.3390/electronics11132012

4.         Yang, Y., Li, T., Wang, Y., Cheng, H., Chang, S. X., Liang, C.,... An, S. (2021). Negative effects of multiple global change factors on soil microbial diversity. Soil biology & biochemistry, 156, 108229. doi: 10.1016/j.soilbio.2021.108229

5.         Yang, Y., Dou, Y., Wang, B., Wang, Y., Liang, C., An, S.,... Kuzyakov, Y. (2022). Increasing contribution of microbial residues to soil organic carbon in grassland restoration chronosequence. Soil Biology and Biochemistry, 170, 108688. doi: https://doi.org/10.1016/j.soilbio.2022.108688

6.         Zhao, Z., Wang, P., Xiong, X., Wang, Y., Zhou, R., Tao, H.,... Xiong, Y. (2022). Environmental risk of multi-year polythene film mulching and its green solution in arid irrigation region. Journal of hazardous materials, 435, 128981. doi: 10.1016/j.jhazmat.2022.128981

Authors need to revise the abstract and conclusion sections. Authors have mentioned numerous results in the conclusion section which should be in the abstract. In conclusion, authors need to mention the findings only rather than quantitative data.

Reviewer 3 Report

this is an interesting manuscript

please see the following comments

1) English language should be improved because right now there are many mistakes. please see some corrections but many more are needed

replace

been fertilized 12 years

with

been fertilized for 12 years

replace

Bio-organic fertilization technology could significantly increase the num- 20
ber and activity of soil microorganisms, and upgrading soil enzymes

with

Bio-organic fertilization technology could significantly increase the num- 20
ber and activity of soil microorganisms, and upgrade soil enzymes

you state

clay minerals were con- 23
nected as nuclei with the capacity to bind carbon,

I dont understand this sentence

you state

Specifically, the O-H has the quickest reaction to aliphatic-C, next by Si-O, Fe-O, and 25
Al-O in M2 treatment I dont understand this sentence. Besides you never explained what M2 treatment was, what O-H means here etc. This should be in the results, not in the abstract

replace

n China, which has 3.2 million hm2 of tea gardens and will produce 2.93 million tons 33
of dried tea in 2022, tea (Camellia sinensis (L.) Kuntze) is a significant commercial crop.

with

n China, which has 3.2 million hm2 of tea gardens and will produce 2.93 million tons 33
of dried tea in 2022, this plant (Camellia sinensis Kuntze) is a significant commercial crop.

replace

The test soil is lateritic red soil, which was collected from the Yingde Test Base of the 99
Institute of Drinking Plants, Guangdong Academy of Agricultural Sciences. The baseis 100
located in Yingde City, Guangdong Province (24°18′ N, 113°23′ E).

with

The test soil was lateritic red soil, collected from the Yingde Test Base of the 99
Institute of Drinking Plants, Guangdong Academy of Agricultural Sciences.
located in Yingde City, Guangdong Province (24°18′ N, 113°23′ E).

replace

It has a subtropical 101
monsoon climate, with an average temperature of 21.1 °C and an average annual precipi- 102
tation of 1906.2 mm. The field experiment was set up in November 2009, with the experi- 103
mental site of 80m × 50m (long × A rectangular plot with a width of), the row spacing of 104
tea trees is 1.5m, a 2m wide isolation row is set between each plot, and a 5m wide protec- 105
tion row is set around the test field.

with

The area is characterized by a subtropical 101
monsoon climate, with an average temperature of 21.1 °C and an average annual precipi- 102
tation of 1906.2 mm (give reference). The field experiment was set up in November 2009, with an experi- 103
mental site of 80m × 50m (length × width), the row spacing between
tea trees was 1.5m, a 2m wide isolation row was set between each plot, and a 5m wide protec- 105
tion row was set around the test field.

replace

until about 1kg of each mixed sample 119
was retained and put into a self-sealing bag to bring it into the room

with

until approximately 1kg of each mixed sample 119
was retained and put into a self-sealing bag until further analysis

etc.

2) there are also mistakes due to carelessness eg in the abstract there is different fornt in some sentences, hm2 2 should be superscipt, CO2 2 should be subscript etc..

3) do not give abbreviations throughout the document that you have not explained eg SOC storage. first give in full and then use only the abbreviations throughout the manuscript

4) there is too much textbook information that should be erased or shortened eg please see lines 15-16, 35-36, 40-42, 43-46, 49-50, 65-70

5) in the introduction you dwell too much on some aspects that you are going to analyze but this is quite confusing to the reader. I suggest that you keep references to minimum and transfer some of the references to the discussion with the aim of comparing your results to theirs, explain more clearly your hypothesis and give a concluding paragraph on why this is important for an international audience

6) in the plot description please give past tense not present or future tense

7) in materials and methods for all apparatuses and reagents used eg sieves, spectrophotometer, colorimeter, slicer, softwares etc give manufacturer, city and country of origin. in some of these OMNIC 9.0 software and Origin 147
9.0 software you give too much information

8) for The method of multi-point mixing, drying weighing method; he total carbon and nitrogen, soluble organic carbon (DOC), activity of nitrite 138
reductase

9) you state The cut samples were irradiated by synchronous radia- 141
tion

in order to analyze what?

10) you state

Jinxuan, an annual tea tree variety, was selected for greenhouse culture

I dont understand this-is this another experiment in a glasshouse? to examine for what? was there also fertilization? it is not understood at all

11) it is impossible to understand what methods were performed on the plants and what on the soil. also in the results you state correlation analysis results but you never described any statistics on your materials and methods. how did correlation analysis happened? was it pearson or spearmans and why? was some other type of correlation? please elaborate

12) In  Table 1. Basic physical and chemical properties of the soil and organic material tested. were these data produced here or were taken from somewhere else? this should be clear

13) in table 2 you should have described the statistics in materials and methods

14) you never described how you calculated yield in the materials and methods

15) I dont understand in your results when you compare between years and when you compare between treatments. i believe in some results you compare years while in others you compare treatments? also if you compare years you should do a repeated measures ANOVA not an one-way ANOVA. please explain in detail

16) I find absolutely no reason to show the equation in table 5 only R2 is needed and out of these it is better to show only the >0.7. how you measured fats, polysacharides and proteins? it is not clear. also how you did the disctinction between Clay, SiO etc.. it is totally baffling

17) you state

he effect of fertilization on soil enzyme activity was 310
statistically significant (P<0.05) in this study

this is very broad statement, when and what enzymes

18) you state According to the infrared spectra in figure 3 and 4 shows that the characteristic peaks 221
(3620 cm1) of clay minerals are from scratch, and the characteristic peaks of macromolec- 222
ular organic substances are either from scratch (such as 1650 cm1 and 2920 cm1) or grad- 223
ually increase in intensity (1080 cm1

i dont understand this sentence at all

I suggest that you explain in a consice manner what you expect to see from the IF method and how is this linked to the comments you make in results and in discussion

19)do not give any comments in the results section. please just explain the results and any comments etc should be given in the discussion section

20) it is absolutely imperative to add a statistics part in you method where you explain exactly what statistical manipulation you did and for what set of data because right now it is impossible to understand where these deductions came from

21) as such discussion should be rewritten

22) for such a long manuscript with so many data the reference list is very small. please add if possible the following papers.

Biosolid‐amended soil enhances defense responses in tomato based on metagenomic profile and expression of pathogenesis‐related genes

Comparative effect of organic, organomineral and
mineral fertilizers on soil properties, nutrient uptake,
growth and yield of maize (Zea Mays)

Use of Biosolids to Enhance Tomato Growth and Tolerance to Fusarium oxysporum f. sp. radicis-lycopersici

Comparative evaluation of changes in soil bio-chemical properties after application of traditional and enriched vermicompost

Vermicompost acts as bio-modulator for plants under stress and non-stress conditions

Microbial activity was greater in soils added with herb residue vermicompost than chemical fertilizer

the references should be at least 50

Round 2

Reviewer 3 Report

The manuscript has been improved. however please still see the following

1) English language corrections are imperative

2) I have suggested the following references which I believe are relevant to the present research but I dont think I can find them in the amended manuscript

Stavridou et al Biosolid‐amended soil enhances defense responses in tomato based on metagenomic profile and expression of pathogenesis‐related genes. Plants

  • Leye Samuel Ayeni, E. Adeleye, J. Adejumo
  • Published 2012
  • Chemistry
  • International Research Journal of Agricultural Science and soil Science Comparative effect of organic, organomineral and

mineral fertilizers on soil properties, nutrient uptake,
growth and yield of maize (Zea Mays)

 Giannakis, I., Manitsas, C., Eleftherohorinos, I. et al. Use of Biosolids to Enhance Tomato Growth and Tolerance to Fusarium oxysporum f. sp. radicis-lycopersici. Environ. Process. 8, 1415–1431 (2021) 

Makkar, C., Singh, J., Parkash, C. et al. Vermicompost acts as bio-modulator for plants under stress and non-stress conditions. Environ Dev Sustain (2022).

Lv, M., Li, J., Zhang, W. et al. Microbial activity was greater in soils added with herb residue vermicompost than chemical fertilizer. Soil Ecol. Lett. 2, 209–219 (2020).

 3) in general I cannot see the answers in all the comments because there is only a new non track changes version uploaded. Please provide a point by point answer to comments and a track changes version as well

4) the references are clearly not written according to instructions for authors

Author Response

Dear reviewer:

I am very grateful to your comments for the manuscript. All of your questions were answered below.

1) English language corrections are imperative

Response 1): Thank you so much for your suggestion. We did correct the grammatical and typographical errors throughout the manuscript.

2) I have suggested the following references which I believe are relevant to the present research but I dont think I can find them in the amended manuscript

Response 2): Thank you so much for your suggestion. We did add the recommended references as follows.

Stavridou et al Biosolid‐amended soil enhances defense responses in tomato based on metagenomic profile and expression of pathogenesis‐related genes. Plants

Sorry, some information is missing and could not find the paper.

    Leye Samuel Ayeni, E. Adeleye, J. Adejumo

    Published 2012

    Chemistry

    International Research Journal of Agricultural Science and soil Science Comparative effect of organic, organomineral and mineral fertilizers on soil properties, nutrient uptake, growth and yield of maize (Zea Mays)

Added in the reference 33

 Giannakis, I., Manitsas, C., Eleftherohorinos, I. et al. Use of Biosolids to Enhance Tomato Growth and Tolerance to Fusarium oxysporum f. sp. radicis-lycopersici. Environ. Process. 8, 1415–1431 (2021) 

Added in the reference 34

Makkar, C., Singh, J., Parkash, C. et al. Vermicompost acts as bio-modulator for plants under stress and non-stress conditions. Environ Dev Sustain (2022).

 Added in the reference 14

Lv, M., Li, J., Zhang, W. et al. Microbial activity was greater in soils added with herb residue vermicompost than chemical fertilizer. Soil Ecol. Lett. 2, 209–219 (2020).

 Added in the reference 45

 3) in general I cannot see the answers in all the comments because there is only a new non track changes version uploaded. Please provide a point by point answer to comments and a track changes version as well

Response 3): I provided the point-by-point response to your comments in the attachment last time. The response to comments of round 1 are as follows.

1) You state “clay minerals were connected as nuclei with the capacity to bind carbon”, I dont understand this sentence. you state “Specifically, the O-H has the quickest reaction to aliphatic-C, next by Si-O, Fe-O, and Al-O in M2 treatment I dont understand this sentence. Besides you never explained what M2 treatment was, what O-H means here etc. This should be in the results, not in the abstract”

Response 1: The formation of the organo-mineral complex is one of the critical factors

that influence the evolution of soil fertility. SR-FTIR analysis revealed that clay minerals were connected as nuclei with the capacity to bind carbon, and that this interaction was aided by organic fertilization. And we correct the description as “the replacement of chemical fertilizer with organic fertilizer can improve the ability of clay minerals and iron/aluminum/silicon oxides to protect aliphatic groups, polysaccharides and proteins”.

2) there are also mistakes due to carelessness eg in the abstract there is different fornt in some sentences, hm2 2 should be superscipt, CO2 2 should be subscript etc..

3) do not give abbreviations throughout the document that you have not explained eg SOC storage. first give in full and then use only the abbreviations throughout the manuscript

4) there is too much textbook information that should be erased or shortened eg please see lines 15-16, 35-36, 40-42, 43-46, 49-50, 65-70

5) in the introduction you dwell too much on some aspects that you are going to analyze but this is quite confusing to the reader. I suggest that you keep references to minimum and transfer some of the references to the discussion with the aim of comparing your results to theirs, explain more clearly your hypothesis and give a concluding paragraph on why this is important for an international audience

6) in the plot description please give past tense not present or future tense

7) in materials and methods for all apparatuses and reagents used eg sieves, spectrophotometer, colorimeter, slicer, softwares etc give manufacturer, city and country of origin. in some of these OMNIC 9.0 software and Origin 147

9.0 software you give too much information

8) for The method of multi-point mixing, drying weighing method; he total carbon and nitrogen, soluble organic carbon (DOC), activity of nitrite 138reductase

Response 2: Thank you so much for your suggestion. We make the necessary corrections in the revised manuscript

9) you state The cut samples were irradiated by synchronous radiation in order to analyze what?

Response 3: Synchrotron-based Fourier transform infrared (SR-FTIR) spectroscopy was used to concurrently characterize physical and chemical organo-mineral interactions using in situ mapping and profiling without the need for exogenous biogeochemistry probes at the submicron scale.

10) you state Jinxuan, an annual tea tree variety, was selected for greenhouse culture. I dont understand this-is this another experiment in a glasshouse? to examine for what? was there also fertilization? it is not understood at all

Response 4: We conducted the pot experiment to test the effect of fertilizers on tea quality and quantity in a glasshouse in Nanjing by using the soil collected from Guangdong.

11) it is impossible to understand what methods were performed on the plants and what on the soil. also in the results you state correlation analysis results but you never described any statistics on your materials and methods. how did correlation analysis happened? was it pearson or spearmans and why? was some other type of correlation? please elaborate

Response 5: The infrared spectra were extracted by OMNIC 9.0 software and the functional groups were analyzed by Origin 9.0 software. For the analysis, SPSS software Version 18.0 for Windows was used for analyzing the data (means ± SD, n = 4) using ANOVA. We used Duncan’s multiple range test at p ≤0.05 to test the differences between different treatments. Pearson’s correlation coefficient (r) values were performed to ex-amine linear correlations at p < 0.05.

12) In Table 1. Basic physical and chemical properties of the soil and organic material tested. were these data produced here or were taken from somewhere else? this should be clear

Response 6: The basic data was offered by Correspondence author Jinchi Tang and also dressed in her published paper. (DOI: 10.13305/j.cnki.jts.2016.01.006), and we added the reference in the revised manuscript.

13) in table 2 you should have described the statistics in materials and methods

Response 7: Thank you so much for your suggestion. We make the necessary corrections in the revised manuscript

14) you never described how you calculated yield in the materials and methods

Response 8: The data of tea quality and quantity was collected from the pot cultivation experiment which was mentioned in 2.2.

15) I dont understand in your results when you compare between years and when you compare between treatments. i believe in some results you compare years while in others you compare treatments? also if you compare years you should do a repeated measures ANOVA not an one-way ANOVA. please explain in detail

16) I find absolutely no reason to show the equation in table 5 only R2 is needed and out of these it is better to show only the >0.7. how you measured fats, polysacharides and proteins? it is not clear. also how you did the disctinction between Clay, SiO etc.. it is totally baffling

Response 9:  For the study, the functional groups were categorized into seven phases based on the stretching frequency (Table4). These recognizable vibrations (v) were clay (3620 cm−1), aliphatic-C-H (2945 cm−1), aromatic-C-C (1650 cm−1), Polysaccharides C-O (1100 cm−1), silicates Si-O (990 cm−1), smectite Al-O (915 cm−1), and iron oxides Fe-O (695 cm−1), respectively (Figure4-5 and Table 4). The colors of functional groups in these photographs ranged from red to blue, according to the relatively strong SR-FTIR absorbance to the

comparatively weak one. The correlation analysis between organic functional groups and minerals in the soil aggregates as influenced by a variety of fertilizer treatments using SR-FTIR showed in Table 5. In addition, the geographical association between clay minerals and organic functional groups is shown in Figure 4-5, in which the absorbance spots in all treatments were collected.

17) you state the effect of fertilization on soil enzyme activity was statistically significant (P<0.05) in this study this is very broad statement, when and what enzymes

18) you state According to the infrared spectra in figure 3 and 4 shows that the characteristic peaks (3620 cm−1) of clay minerals are from scratch, and the characteristic peaks of macromolecular organic substances are either from scratch (such as 1650 cm−1 and 2920 cm−1) or gradually increase in intensity (1080 cm−1

i dont understand this sentence at all

I suggest that you explain in a consice manner what you expect to see from the IF method and how is this linked to the comments you make in results and in discussion

Response 18:  All the information that reviewer mentioned were presented in 3.5. and response 9.

19)do not give any comments in the results section. please just explain the results and any comments etc should be given in the discussion section

20) it is absolutely imperative to add a statistics part in you method where you explain exactly what statistical manipulation you did and for what set of data because right now it is impossible to understand where these deductions came from

21) as such discussion should be rewritten

22) for such a long manuscript with so many data the reference list is very small. please add if possible the following papers.

Response 18: Thank you so much for your suggestion. we make the necessary corrections in the revised manuscript and the references added to 48

Last but not the least, all the specific comments were revised in the manuscript.

Thank you and all the reviewers for the kind advice.
